# OpenReview forum: "An Emulator for Fine-tuning Large Language Models using Small Language Models"
_ICLR.cc/2024/Conference — ICLR 2024 poster_

### Official Review · Reviewer_fHwV · 2023-10-28

**Soundness:** 4 excellent
**Presentation:** 2 fair
**Contribution:** 4 excellent
**Rating:** 8
**Confidence:** 5

**Summary:**

This paper proposes a method, emulated fine-tuning (EFT), to decouple the effect of the knowledge learned by a large model during pre-training and the knowledge learned by a small model during fine-tuning (or vice versa). Given a smaller pre-trained model $\pi_{ref}^{M}$, its fine-tuned counterpart $\pi^{M}$, and a larger pre-trained model $\pi_{ref}^{N}$, EFT emulate the effect of combining the pre-trained knowledge in $\pi_{ref}^{N}$ and the knowledge learned by $\pi^{M}$ by simply computing
$\log \pi_{ref}^{N}(y|x) + (   \log \pi^{M}(y|x)     - log  \pi_{ref}^{M}(y|x) )$.

Using EFT, they show that scaling pre-training (model size) mostly improves factuality, and scaling fine-tuning (model size) mostly improves helpfulness.
They also propose a special case of EFT that uses a large pre-trained model and two small models (one pre-trained and the other fine-tuned) called up-scaling, which can emulate the case when fine-tuning the large model without actually fine-tuning the large model.
They show that EFT can also be combined with speculative decoding to speed up the inference speed, and they use top-p truncation to improve the performance of up-scaling.

**Strengths:**

- EFT is motivated by prior observations that LLMs can be seen as a reward model. They use this observation to decouple the effect of the model scale for pre-training and fine-tuning. This is a very interesting interpretation.
- EFT is very simple to implement, and up-scaling can improve the pre-trained LLMs' performance without fine-tuning the large LLMs.
- The takeaway of the paper is very interesting.
- The evaluation of the paper is solid: they use GPT-4 evaluation and conduct further human evaluation to justify the validity if using GPT-4 as the evaluator.
- The paper is mostly easy to follow. The takeaways are clear and the experiment settings are clear too.

**Weaknesses:**

- EFT requires three models during inference. Three models occupy a lot of space and lengthen the inference time. Still, the inference time can be reduced by speculative decoding. This makes me doubt the practical value of EFT and upscaling.
- It is unclear how much up-scaling recovers the performance of fine-tuning the large model. The paper only reports the improvement of the up-scaled model compared with the pre-trained large model, but I think it is also important to report the performance of the fine-tuned large model. This way we can understand if up-scaling can close the gap of fine-tuning the large model, or if the performance still largely lags behind the fine-tuned large model. This is an important information for practical use cases. Still, I want to emphasize that this paper's contribution to proposing the EFT framework is ample contribution, and even if there are several drawbacks in practical usage, I still consider this paper a good paper.
- The math in Section 3 is kind of confusing. I will elaborate on them in the question part.
- Some minor presentation issues. The paper might need proofreading.
   -  Section 3.1:  ` we view the result of fine-tuning is the solution to` $\to$ `we view the result of fine-tuning **as** the solution to`.
   -  Section 3.1: `Crucially, while the EFT framework is justified with an RL interpretation is applicable to any fine-tuned model` $\to$ `Crucially, while the EFT framework is justified with an RL interpretation, **it** is applicable to any fine-tuned model`
   - Page 6: `with on` $\to$ `with`
   - Section 4: `While prompts in the HH dataset are more everyday and conversational, asking for movie recommendations or instructions or home maintanence tasks.` $\to$ This is not a complete sentence.
    - Section 4.3 and 4.4: I think you should first introduce the method and then refer to the experiment results in the tables.

**Questions:**

1. I am confused about the math in Section 3.
  - 1.1 Why is the $r_{\pi}(x,y)$ on Line 6 in page 4 $\beta \log \frac{\pi_{ft}(y|x)}{\pi_{ref}(y|x)}$ instead of $\beta \log \frac{\pi_{ft}(y|x)}{\pi_{ref}(y|x)} + \beta \log Z(x)$?
  - 1.2 In Line 6,  $r_{\pi}(x,y) = \beta \log \frac{\pi_{ft}(y|x)}{\pi_{ref}(y|x)}$. But in the last part of Equation (3) and in the following texts, it seems that  $r_{\pi}(x,y) = \log \frac{\pi_{ft}(y|x)}{\pi_{ref}(y|x)}$. Why is there such a difference?

2. The paper mainly focuses on generation tasks related to factuality and helpfulness/harmlessness. I wonder does EFL, or precisely, up-scaling, also show improvement in multiple-choice datasets like MMLU?

---

> ### Author Response · Authors · 2023-11-19
>
> Thank you very much for your in-depth review of our work- we're glad you enjoyed the paper!
>
> **Q: Space and time costs of EFT**
>
> **A:** While EFT up-scaling does require two extra models, these two models can be significantly smaller; for example, upscaling Falcon 7B to 180B adds fewer than 8% of the total parameters. While the inference cost is slower, as you point out, speculative decoding is very effective here; further, the primary benefit of EFT is that fine-tuning can be skipped altogether. We have clarified this aspect of the contribution of EFT, that it eliminates the cost of the fine-tuning stage (rather than speeding up inference) in the end of the introduction.
>
> **Q: Comparing with the large fine-tuned model**
>
> **A:** To clarify our results, our experiments do use the fine-tuned large model (not the fine-tuned base model) as the point of reference. We have clarified this important point in the experiments section of the revised paper!
>
> **Q: Typos and experiment section structure**
>
> **A:** Thank you very much for pointing these issues out- the typos have been fixed, and the experimental sections have been re-structured according to your feedback in the revised paper.
>
> **Q: Regarding the reward function and the partition function Z(x)**
>
> **A:** For the special case of the reward being defined as the log ratio of a policy and the same reference model used in the RL problem (which we can assume to be the case), the partition function is simply 1, so the log Z(x) term is zero. While this is alluded to in footnote 1, we have revised the section to make this point clearer.
>
> In the general case, while the partition function is indeed part of the reward function, the partition function Z(x) is still a constant with respect to different possible responses/next tokens y; therefore it does not change the optimal policy. In the case of the per-token approximation that is actually used to sample from an EFT model in practice, this partition function is approximated at the per-token level (it is the normalizer for the softmax over the next token after doing the EFT logit arithmetic).
>
> **Q: Beta in the reward function**
>
> **A:** When viewing a fine-tuned model as the result of KL-constrained RL, a large value of beta corresponds to increasing the scale of the reward function, but using a correspondingly stronger KL constraint; a small value of beta correspond to squashing the reward function, but using a correspondingly weaker KL constraint. Thus in the specific case where we are simply using the RLHF framework to interpret a fine-tuned model (i.e., we have fixed the optimal policy [which is the fine-tuned model] and the reference model and are solving for the reward), the value of beta is simply an extra degree of freedom. For this reason we simply assume beta=1 for simplicity for the remainder of the paper. Thank you for raising this question- we have made this decision more explicit in the revised version of Section 3.1.
>
> **Q: Evaluating EFT on other datasets**
>
> **A:** We have added a new set of experiments on the code generation setting, which can be found in the newly-added Figure 7. We find that EFT shows very strong performance in the code generation setting as well. We use the code generation benchmark HumanEval and find the pass@1 solve rates to be:
>
> | Model | pass@1 solve rate |
> | --- | --- |
> | Llama-2-7b-chat | **0.128** |
> | EFT 7b->70b (beta=1.5) | **0.256** |
> | Llama-2-70b-chat | **0.280** |
>
> EFT up-scaling recovers ~84% of the solve rate improvement of 70b-chat compare to 7b-chat, without actually doing any fine-tuning.

---

> > ### Comment · Reviewer_fHwV · 2023-11-21
> >
> > Thank you for your responses. They clarify all my questions.
> > After reading the reviews from fellow reviewers and the author responses, I keep my original evaluation: **This is a good paper and should be accepted**.

---

### Official Review · Reviewer_oV1U · 2023-10-29

**Soundness:** 3 good
**Presentation:** 2 fair
**Contribution:** 3 good
**Rating:** 6
**Confidence:** 2

**Summary:**

The paper introduces emulated fine-tuning (EFT) to combine the knowledge gained from pre-training and the knowledge gained from fine-tuning from different scales and provide mathematical intuition.
The experiment shows a larger fine-tuning scale improves helpfulness and a larger pre-training scale improves factuality.
The method leads to a resource-efficient fine-tuning method, combining a large pre-training scale with a small fine-tuning scale.

**Strengths:**

(1) The idea of pre-training and fine-tuning at different scales is very interesting. (contribution (a))

(2) The paper shows that a larger fine-tuning scale improves helpfulness and a larger pre-training scale improves factuality as Figures 3 and 4, which potentially provide intuition to guide fine-tuning methods. (contribution (b))

(3) The proposed method provides a testing-time flexibility to the trade-off of helpfulness and harmlessness.

**Weaknesses:**

(1) In contribution (c), the paper kind of implies that the up-scaling is beneficial to efficiency, but what's the performance/inference cost comparison between inferencing the small model twice and Parameter-Efficient Fine-Tuning (PEFT) such as LoRA? If the proposed method cannot approach the performance/inference cost of PEFT, then it's hard to say it's more efficient.

(2) While the proposed method is evaluated with varied ablation studies, it seems that the method is not to be rigorously compared with directly fine-tuning the pre-trained model (combining different scales may or may not cause trouble to performance). If the proposed method cannot match the performance of directly fine-tuning the pre-trained model, then the study of contribution (b) would be less meaningful.

**Questions:**

I like the proposed idea very much, but I am concerned about the contribution of the paper.

Performance-wise it's not clear whether combining different scales will lead to performance degradation. Efficiency-wise it's not clear whether it's better than Parameter-Efficient Fine-Tuning (PEFT). I understand the method provides a trade-off between helpfulness and harmlessness, and a method to explore the effects of pre-training and fine-tuning, but I think it's only meaningful when the performance is not degraded after combining different scales.

---

> ### Author Response · Authors · 2023-11-19
>
> Thank you very much for your review and feedback- we're glad you found the contributions interesting!
>
> **Q: Inference cost of EFT vs LoRA**
>
> **A:** In general, the practical benefit of EFT (in addition to the scientific benefit of studying pre-training and fine-tuning independently) is the ability to avoid fine-tuning altogether, rather than improved inference performance. We have clarified this aspect of the contribution of EFT in the end of the introduction. LoRA inference performance should be the same as the pre-LoRA model, as the LoRA parameters can be merged with the pre-trained model parameters, leaving the forward pass unchanged. Thus, our decoding efficiency evaluation in Table 1 shows that naive EFT decoding generates about 33% fewer tokens per second than a plain model (or equivalently a 70B LoRA model). However, with speculative decoupled decoding, we can speed up EFT decoding by a factor of at least 2.5x.
>
> **Q: Comparing to actually fine-tuning**
>
> **A:** To clarify our experiments, **the main results in Figures 3 and 4 do use the large fine-tuned model as the point of reference.** That is, the x axis is normalized so that x=0 corresponds to the performance of the *small fine-tuned model* (e.g., Llama-2-7b-chat), and x=1 corresponds to the performance of the *large fine-tuned model* (e.g., Llama-2-7b-chat). Figure 4 therefore shows that, for example, *up-scaling Llama-2-7b-chat to 70b achieves equal or better factuality than Llama-2-70b-chat on Anthropic prompts, without actually fine-tuning*, as the factuality score is 1.04 (meaning that EFT produces 104% of the factuality improvement from 7b-chat to 70b-chat). While EFT does not typically recover all of the performance of fine-tuning the larger base model (that is, the bars don't typically reach all the way to one), EFT does achieve a large amount of the gains that actual fine-tuning would have, but crucially, EFT does not require any fine-tuning at all.

---

> > ### Comment · Reviewer_oV1U · 2023-11-22
> >
> > Thank you for your response to my questions. I raised my score to 6. I am concerned about the performance. I would expect some performance drop when we combine models from two scales. Is it so easy to align the predictions of models from two scales?

---

> > > ### Author Response · Authors · 2023-11-22
> > >
> > > We appreciate the engagement and are glad our response was helpful.
> > >
> > > Whether we see a performance drop when combining two models depends on the baseline we compare with. To take the newly-added Figure 7 as an example, when we combine a small fine-tuned model and large base model (Llama-2-**7b-chat** + Llama-2-**70b-base**), the resulting EFT model is **stronger than either constituent model Llama-2-7b-**chat** or Llama-2-70b-**base****.
> > >
> > > Comparing to Llama-2-**70b-chat** (essentially the upper bound on performance), in this case the EFT model recovers 84% of the improvement of the 70b **chat** model over the 7b **chat** model; thus, as you suggest, EFT may not always recover *all* of the performance that we would see *if we were able to fine-tune the large model*, but it might be quite close. In the case of computationally-constrained environments where fine-tuning the large model is infeasible, EFT may allow us to reap the vast majority of the gains of fine-tuning the large model, without needing to do any fine-tuning at all.
> > >
> > > In terms of the ease of aligning predictions of models from different scales, we interpret our results as showing the 'behavior delta' learned at one scale is indeed 'compatible' with the base model predictions at other scales, enabling nearly lossless 'grafting' of knowledge in some cases. In other words, base models of different sizes produce similar enough conditional distributions that this behavior delta points in the right (distributional) direction, more or less regardless of base model size.

---

### Official Review · Reviewer_UMGS · 2023-10-31

**Soundness:** 3 good
**Presentation:** 2 fair
**Contribution:** 3 good
**Rating:** 6
**Confidence:** 3

**Summary:**

The paper proposes to decouple the fine-tuning and pre-training in an LLM by the reinforcement learning theory. By specifying the enhancement from fine-tuning based on the pre-training, it is possible to introduce the enhancement of smaller models to the larger ones so as to reduce the commuting cost of fine-tuning a large model. This approach is called emulated fine-tuning by the authors. The idea of EFT is tested with different LLM families and evaluated by GPT-4 by measuring the harmfulness, helpfulness, and factuality.

**Strengths:**

The paper provides a theoretical explanation of a simple framework that can greatly reduce the computing cost of pre-training large language models. By incorporating the pre-training of smaller models, we can use the EFT to get an enhanced performance with larger models. The enhancement is evaluated by GPT-4 by measuring the harmfulness, helpfulness, and factuality. The reinforcement learning theory is convincing and clear to demonstrate the effectiveness of the proposed method.

**Weaknesses:**

1. The evaluation is limited. Fine-tuning is widely used and not only limited to obtaining a chatbot. More tasks can be used to verify the idea of EFT, such as code generation, question-answering, etc.
2. Though GPT-4 is widely used as the judge to tell the performance of LLMs, more objective metrics can also be used to evaluate the LLMs.

**Questions:**

1. In Figure 3, the values in the chart are labeled but in Figure 4, the values are not. It would be helpful to know the explicit values in the charts.
2. As mentioned in Section 4: Models, three separate families of pre-trained language models are used. According to the theory introduced in Section 3, it is also possible to verify the idea across different families of PLMs. For example, what will the performance be when incorporating the knowledge learned by fine-tuning Llama-7b to Falcon-180?

---

> ### Author Response · Authors · 2023-11-19
>
> Thank you very much for your review! To address your questions:
>
> **Q: Evaluations beyond question-answering/dialogue**
>
> **A:** Thank you for this suggestion! We performed a code generation experiment in the newly-added Section 4.4. We find that EFT provides very strong performance for code generation. We use the code generation benchmark HumanEval; we find the pass@1 solve rates to be:
> Llama-2-7b-chat: 0.128
> EFT 7b->70b (beta=1.5): 0.256
> Llama-2-70b-chat: 0.280
> That is, EFT up-scaling recovers ~84% of the solve rate improvement, without actually doing fine-tuning.
>
> **Q: GPT-4 evaluations**
>
> **A:** Our new experiment on code generation does not use GPT-4 to perform evaluation; rather, it executes the generated code and checks to see if it passes a suite of test cases. In this case, where we have access to "gold standard" evaluations, EFT still shows very strong performance.
>
> **Q: Exact values for Fig 3**
>
> **A:** We have revised the per-dataset and per-model charts to include the exact values (and compressed these to a single figure). Please refer to the updated Figure 4.
>
> **Q: Combining models from different families**
>
> **A:** You are correct that EFT makes no assumption that the models being mixed come from the same family. However, as a practical consideration, applying EFT is much easier when the models share the same tokenizer (since the EFT re-weighted log probabilities are renormalized on a per-token basis). We performed an experiment up-scaling Llama-2-7b-chat with Llama-1-65b-base, using the logits Llama-1-65b-base + (Llama-2-7b-chat - Llama-2-7b-base); these models share the same tokenization, making the implementation straightforward. We find that up-scaling this way improves solve rate from 0.128 to 0.17. This improvement is substantial, though it is smaller than the improvement from up-scaling with Llama-2. The discrepancy is likely due to a combination of Llama-1 being weaker at coding and some slippage in performance due to the Llama-2 behavior delta being best matched to Llama-2 base models. Further exploration of merging and stitching the knowledge of models from different families is a very interesting and exciting direction for future work!

---

### Official Review · Reviewer_XUbA · 2023-11-02

**Soundness:** 3 good
**Presentation:** 2 fair
**Contribution:** 3 good
**Rating:** 6
**Confidence:** 2

**Summary:**

This paper proposes a sampling method to evaluate the effects of different scales of pre-training and fine-tuning, which proves larger pre-training and small fine-tuning datasets are better. Also, it provides an ensembling strategy for different models at different scales, which seems inspiring that the up-scaling technique can approximate the compute-intensive result of large models without extra resources.

**Strengths:**

1. The proposed emulated fine-tuning framework can approximate the results without associated computational expense, and prove the helpfulness and factuality of each procedure.
2. The up-scaling technique can approximate the results of compute-intensive models, however, without the associated computational expense.

**Weaknesses:**

1. The paper is not easy to follow. More important details are needed for understanding and reproduction. For EFT, how the sampling is conducted?  Does different scales affect the sampling strategy? For the ensembling, how the new weights are obtained?
2. It claims that up-scaling can approximate the results of compute-intensive models, which need more experiments and comparison.

**Questions:**

Please refer to the weakness part.

---

> ### Author Response · Authors · 2023-11-19
>
> Thank you very much for your review and feedback! To address your questions:
>
> **Q: How is sampling conducted?**
>
> **A:** The EFT sampling procedure is defined in Eq. 5, in the paragraph titled *"Sampling with Emulated Fine-tuning."* For example, when we emulate fine-tuning llama-70b-base using llama-7b-chat, we combine the logits from llama-7b-chat, llama-7b-base, and llama-70b-base as: `eft_logits = llama-70b-base_logits + (llama-7b-chat_logits - llama-7b-base_logits).` See also Figure 2 for a diagram of sampling.
>
> **Q: Does different scales affect the sampling strategy?**
>
> **A:** The sampling strategy is the same for all models and scales.
>
> **Q: For the ensembling, how the new weights are obtained?**
>
> **A:** In our model-mixing experiments, no new model weights are obtained; the mixing occurs purely at the sampling step, where the output logits of the two models are combined.
>
> We will revise section 3.2 of the paper to make each of these points more explicit and clear!
>
> **Q: It claims that up-scaling can approximate the results of compute-intensive models, which need more experiments and comparison.**
>
> **A:** We have included additional experiments with up-scaling in the newly-added Section 4.4, in Figure 7. We conduct an experiment on the code generation benchmark HumanEval; we find the pass@1 solve rates to be:
>
> | Model | pass@1 solve rate |
> | --- | --- |
> | Llama-2-7b-chat | **0.128** |
> | EFT 7b->70b (beta=1.5) | **0.256** |
> | Llama-2-70b-chat | **0.280** |
>
> That is, EFT up-scaling recovers ~84% of the solve rate improvement, without actually doing fine-tuning. Upscaling Llama-2-7b-chat on ELI5 can also recover nearly all of the factuality improvement of Llama-2-70b-chat compared to the smaller Llama-2-7b-chat in ELI5 questions (see Figure 7-right).

---

### Author Response · Authors · 2023-11-19
**General response and new results**

We very much appreciate all of the reviewers' feedback! We wanted to highlight one newly-added result and one clarification that came up a few times:

**Newly-added result: We conducted experiments in code generation to complement our dialogue and question-answering experiments. EFT shows strong performance in this setting as well (see newly-added Figure 7).** We compare Llama-2-7b-chat, EFT upscaling of Llama-2-7b-chat to 70b, and Llama-2-70b-chat on the HumanEval code generation benchmark. We find solve rates of:

| Model | pass@1 solve rate |
| --- | --- |
| Llama-2-7b-chat | **0.128** |
| EFT 7b->70b (beta=1.5) | **0.256** |
| Llama-2-70b-chat | **0.280** |

That is, EFT up-scaling recovers ~84% of the solve rate improvement, without actually doing fine-tuning. EFT's strong performance in this setting is additional evidence of the general applicability of the framework.

**Clarification: Our current results do use the *true fine-tuned large model* as the reference for comparison, not the large pre-trained model.** That is, for the results in Figures 3 and 4, a score of x=0 corresponds to the performance of the small fine-tuned model (e.g., Llama-2-7b-chat); a score of x=1 corresponds to the performance of the small fine-tuned model (e.g., Llama-2-70b-chat). Therefore, our results suggest that EFT indeed captures a large amount of the improvement seen from actually fine-tuning the large model.

---

### Meta-Review · Area_Chair_NuUo · 2023-12-14

**Metareview:**

This paper proposes a method to utilize small language models for finetuning LLMs, called emulated fine-tuning using ideas for reinforcement learning and a sampling strategy. They also propose up-scaling which avoids finetuning LLMs altogether through ensembling large and small models. Results are presented with a family of LLMs and finetuned small models show that both pretraining and finetuning offer complementary benefits, and upscaling results in benefits similar to finetuning without necessitating parameter updates.

Strengths: The methods in the paper are a great step forward in efficient language modeling, and the accompanied theoretical principles are an added strength of the paper. The paper presents strong empiricism to support its claim.

Weaknesses: Some details on efficiency, method parameters, reward functions and experimental settings were omitted in the original paper. However, after discussion with reviewers, authors provided these details mostly to the satisfaction of the reviewers. The draft was somewhat hard to follow for some reviewers.

**Justification For Why Not Higher Score:**

See weaknesses above; there were many clarifications that the reviewers asked for, which were not provided in the original paper.

**Justification For Why Not Lower Score:**

This paper merits acceptance for the interesting findings on efficient language model adaptation.

---

### Decision · Program_Chairs · 2024-01-16

Accept (poster)